# Designing and evaluating a public engagement activity about sea level rise

Nieske Vergunst[1], Tugce Varol[2], Erik van Sebille[1]

[1]Freudenthal Institute, Utrecht University, Utrecht, 3584 CC, the Netherlands

[2]Maastricht University, Maastricht, 6211 LK, the Netherlands

*Correspondence to*: Nieske Vergunst (N.L.Vergunst@uu.nl)

**Abstract.** In this paper, we describe the design process of a public engagement activity about sea level rise aimed at young adults aged 16 to 25 living in the Netherlands, intended to reduce participants' psychological distance to sea level rise. We conducted the activity at multiple occasions, including a science festival and vocational education classrooms, and performed

a statistical analysis of the impact measurement among 117 participants. Based on the analysis and observations, we conclude that the activity resonated well with our target audience, regardless of their level of science capital. We suggest that a design study approach is well-suited for the development of similar activities, and recommend a focus on personal relevance, interactivity, and accessibility in public engagement activities. While the game resonated well with participants, the impact may vary in different educational or cultural contexts, particularly where engagement with sea level rise is low.

## 1 Introduction

Engagement between science and society, or *public engagement*, allows science to achieve more transparency and societal impact (Boon et al., 2022). Studying the impact of public engagement activities helps academics and science communicators make informed decisions about allocating their resources and enhancing the efficacy of their public engagement activities (Moser, 2009; Stilgoe et al., 2014).

This study focused on a public engagement activity concerning sea level rise, a consequence of climate change that has worldwide consequences, but is specifically relevant for the Netherlands, where 59% of the country's land surface is prone or sensitive to flooding (PBL, 2010). In the subsequent sections, we discuss the design, implementation, and evaluation of this public engagement activity, which we have titled Sea level game 2080.

### 1.1 Climate change and sea level rise

Global surface temperatures are already 1.35°C higher than in the last half of the 19th century (Lindsay & Dahlman, 2024), and continue to rise (IPCC, 2023). The global sea level is rising at an increasing rate: 1.3 mm per year between 1901 and 1971, 1.9 mm per year between 1971 and 2006, and 3.7 mm per year between 2006 and 2018 (ibid.). Worldwide sea levels have already risen by about 20 cm over the course of the 20th century; the next 20 cm of sea level rise will most likely be reached

between 2025 and 2070, and another 20 cm on top of that between 2050 and 2100 (Le Cozannet et al., 2022). This is expected to cause not only increased chances of flooding, both from the sea and from rivers, but also a myriad of other problems, such as soil salinization and damage to wooden house foundations (KNMI, 2015; Wolters et al., 2018; IPCC, 2023).

In this study, we focus on audiences living in the Netherlands. In a 2020 poll in the Netherlands, 72% of participants indicated that they are worried about climate change in general (Kaal & Damhuis, 2020). Since there are no recent studies on attitudes towards sea level rise in the Netherlands, we draw inspiration from studies in Australia, New Zealand, the United States, and the United Kingdom. A 2016 study done in Australia indicated that participants tend to view climate change as 'psychologically distant' (Jones et al., 2016): distant in time (i.e., far in the future), socially distant (i.e., happening to other people), geographically distant (i.e., happening far away), and uncertain. A similar study in New Zealand showed that participants tend to have rather accurate ideas of current sea level rise predictions, but when asked about worst-case scenarios, they strongly overestimate what is seen as scientifically plausible (Priestley et al., 2021). About half of Americans (52%) are worried about rising sea levels, which is quite a lot less than the amount of people worried about droughts (75%), extreme heat (74%) and water shortages (72%) (Leiserowitz et al., 2023). And in a study in the UK, about two-thirds of participants indicated that they were concerned about sea level rise, while only about one-third saw themselves as well-informed on the topic (Chilvers, 2014).

## 1.2 Climate communication and public engagement

Climate communications aim to achieve multiple objectives, such as educating audiences outside academia on various aspects of climate change, or changing their attitudes and behaviors (Besley & Dudo, 2017). However, communicating climate change is challenging for several reasons, including a large diversity in audiences (Illingworth, 2023), a general shallow understanding of climate change, and growing feelings of overwhelm and hopelessness (Moser, 2016).

Climate communication often has the goal of changing people's behavior and making them act more climate-consciously. However, the effectiveness of climate communication tends to be hampered by various factors that may impede people's willingness to change their behavior, such as perceived social inaction and the inadequacy or unattractiveness of more climate-conscious options (Whitmarsh et al., 2013). In some cases, giving people more insight into the climate consequences of their own behavior might even decrease their willingness to take more climate-conscious actions (ibid.). On the other hand, there are various factors that may increase people's willingness to exhibit more climate-conscious behaviors, for instance, putting emphasis on personal responsibility (Bouman et al., 2020), taking a positive and motivational approach, putting emphasis on agency and possible actions of individuals (Whitmarsh et al., 2013), and introducing a sense of urgency by framing a communication effort such that it lowers psychological distance to climate change (Spence et al., 2011).

Serious games have emerged as effective tools for engaging audiences in climate adaptation challenges, offering interactive experiences that can enhance understanding and foster decision-making skills for complex issues such as sea level rise and coastal adaptation (Flood et al., 2018). Lawrence and Haasnoot (2017) underscore the importance of such games in facilitating adaptive pathways planning in the face of uncertainty in climate change, while Yang et al. (2024) provide a gameplay analysis that highlights serious games as tools for climate adaptation learning.

Building on these studies, the Sea level game 2080 extends the application of serious games by specifically targeting young adults' perceptions of personal responsibility in addressing sea level rise. Unlike previous games focused on adaptive decision-making, this game employs a dilemma-based approach to reduce psychological distance in sea level rise awareness.

## 1.3 Science capital

Public engagement efforts of academics and science communicators tend to focus on groups that are relatively close to academia (Canfield et al., 2020). In the context of climate change, Kaal & Damhuis (2020) observe that higher educated people (sic) feel a higher responsibility to combat climate change than lower educated people. However, given the pervasive global effects of climate change, it might be crucial to direct efforts on reaching audiences that are less familiar with science as well, to effectively address this issue. *Science capital* describes a person's views about and familiarity with science, including their knowledge, attitude, experiences and skills (Archer et al., 2015; Peeters et al., 2022).

## 1.4 Measuring impact of public engagement

The impact of climate communication activities by scientists is rarely evaluated (Wijnen et al., 2023). However, such impact evaluations can help foster critical reflection on the quality and effectiveness of such activities and offer essential practical insights for public engagement practitioners (Jensen, 2015, Strick & Helfferich, 2023).

For evaluating the impact of our public engagement activity, we used the methods and tools provided by IMPACTLAB (Land-Zandstra et al., 2023). *Impact* is the term generally used to describe long-term effects on society, which is hard to measure. The IMPACTLAB tools were designed to give an indication of the impact of a public engagement activity by measuring its *output* (e.g., quantifying results such as event attendance) and *outcomes* (e.g., changes in knowledge, attitude and/or behavior of participants). Although we use the word 'impact' in the evaluation, we are referring to the 'outputs' and 'outcomes' of our public engagement activity.

## 1.5 Research objectives

In this study, we describe the design of a public engagement activity about climate change, specifically sea level rise, that was targeted to young adult audiences with a broad range of science capital. The aim of the activity was to decrease the participants' psychological distance towards sea level rise. We included an impact assessment to find out whether the activity had a positive impact on participants and to find out if the science capital was a predictor of the impact outcomes. We describe the public engagement activity in Section 2 of this paper. In Section 3, we describe the design study approach, consisting of various design phases, that we used to conceive and develop the activity.

In Section 4, we describe the quantitative analysis of the impact measurement questionnaire, which was filled out by participants after completing the activity. We measured subjective (self-reported) outcomes, following Strick & Helfferich (2023). We supplemented the quantitative data with some qualitative observational data. We describe the results of the analysis in Section 5. In Section 6, we present our conclusions and discuss the limitations of our study and directions for future research.

## 2. Sea level game 2080

In this section, we introduce the final version of the Sea level game 2080, before describing the design process in Section 3.

A schematic representation of the gameplay setup is presented in Fig. 1. Two identical game boards (Fig. 2) are placed on either side of a set of large computer screens facing towards the game boards; each screen shows a Powerpoint presentation, controlled from a laptop. (The game is also playable in a 'paper' version, where the screens are replaced by document stands with printouts.) The participants are split up into two teams of one to five players each, Team Solution level and Team Sea level. The players gathered around their respective game boards, facing the computer screens. The game leader gives the teams

a brief introduction about the game. Each participant chooses a playing piece and places it on the 'start' section of the game board.

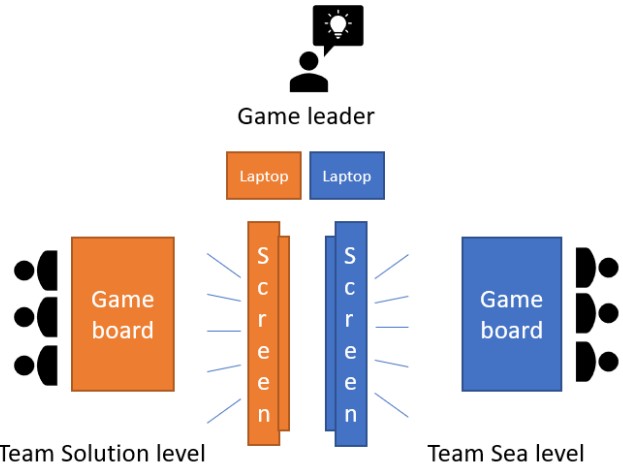

**Figure 1: The Sea level game 2080 is played by two teams, each with their own game board, on either side of a pair of screens. The**
**screens each show their own set of dilemmas, controlled by a game leader, who also leads the introduction and discussion phases of the game.**

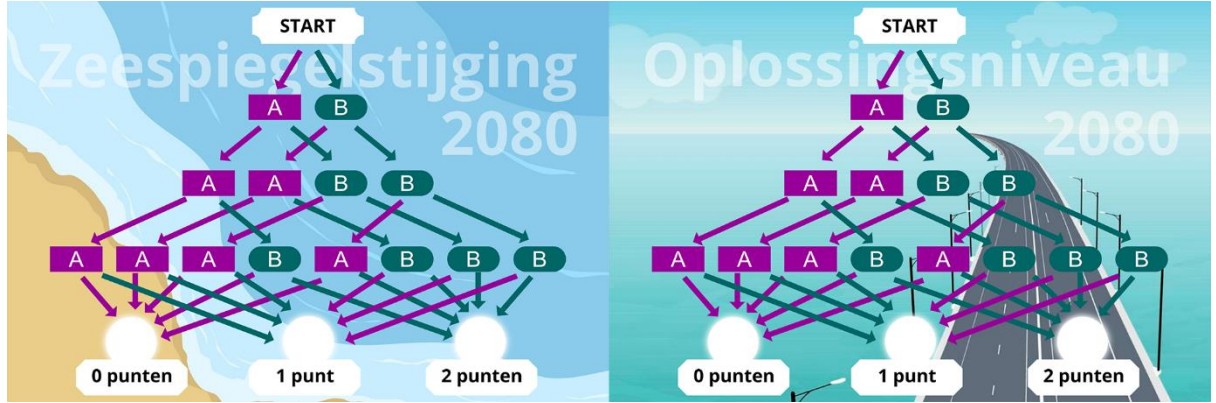

**Figure 2: On the left, the game board for Team Sea level; on the right, the game board for Team Solution level. On both game boards, players move their playing pieces based on the choices they make, and end up in one of three outcomes.**

Then, each team is presented with their first dilemma, displayed on the computer screen. The dilemmas relate to each team's own 'aspect' of sea level rise: the dilemmas for Team Sea level determine the amount of sea level rise; the dilemmas for Team Solution level determine the extent to which mitigating solutions are implemented for problems caused by rising sea levels. The complete set of dilemmas is included in the supplementary materials, but to illustrate the nature of the dilemmas, we present one question given to Team Sea level:

*Question 1. One of your friends is organising a day out to the beach for your entire group of friends. They have come up with two options and are asking everyone to vote. Which one will you choose?*

*A. A beach walk where we clean up plastic, and then a cooking workshop with home-grown vegetables or an organic beer-making workshop.*

*B. We go jet skiing at sea with the whole group, then drive to a beach club for a delicious barbecue.*

The players read the dilemmas on the computer screen and make their choices. Some discussion within the teams is allowed, but participants move their own playing pieces according to their individual choices. The game leader waits for all players to move their playing pieces before moving on to the next dilemma, so both teams go through the dilemmas at the same pace, and finish at the same time.

After the teams have gone through the four dilemmas, the playing pieces end up at one of the three results at the bottom of the

game board, and the game leader tallies up the scores and uses a scoring table to determine the outcome for each team: low, medium or high. For Team Sea level, the three outcomes refer to a low, medium or high amount of sea level rise; for Team Solution level, the three outcomes similarly refer to a low, medium or high amount of mitigation measures being taken. The game leader combines the scores for the two teams, leading to one of nine future scenarios (see Fig. 3). The game leader reads the scenario out loud to the participants, and asks them to reflect on the future that has resulted from their choices: did they

expect this outcome and how do they feel about it?

| SEA LEVEL RISE → <br> SOLUTION LEVEL ↓ | LOW | MEDIUM | HIGH |
|---|---|---|---|
| **LOW** | LOW/LOW | LOW/MEDIUM | LOW/HIGH |
| **MEDIUM** | MEDIUM/LOW | MEDIUM/MEDIUM | MEDIUM/HIGH |
| **HIGH** | HIGH/LOW | HIGH/MEDIUM | HIGH/HIGH |

**Figure 3: The table with outcomes shows how the final scores are combined into one future scenario: for both sea level rise and solution level, the teams end up in either 'low', 'medium' or 'high', leading to one of nine scenarios. For instance, in the top right scenario, the sea level is high, while only a low amount of mitigating measures have been taken.**

The complete set of future scenarios is included in the supplementary materials, but as an illustration, the scenario for 'high sea level rise, low solution level' is as follows:

*The year is 2080. Sea levels have risen more than we would like, and are rising faster and faster. And yet our dikes and flood defenses are not well maintained. In the cities, quays sometimes collapse and across the country, cars, trains and boats are often stuck in traffic jams at broken locks and bridges. Due to the risk of flooding, banks no longer give mortgages for houses along the major rivers and on the coast. Due to the higher and saltier groundwater, houses are more likely to suffer from rotting foundations, trees are blown over due to rotten roots, and farmers are starting to experiment with growing seaweed and other saltwater plants. We welcome climate refugees in the Netherlands from warm, dry countries. But how long can we stay here ourselves?*

After a few minutes of discussion and reflection, the participants are asked to fill out an impact measurement questionnaire: a digital or paper questionnaire, consisting of an explanation of the study and a disclaimer, and six questions:

- Age;
- Two representative questions on science capital in general (following Land et al., 2023);
- One question on science capital about sea level rise specifically;
- A multiple-choice question to measure response efficacy; and
- A multiple-choice question to measure perceived relevance.

All versions of the Sea level game 2080, including the various prototypes and the final version, can be found in the supplementary materials.

## 3. Design process

The design process, as used in Veldkamp et al. (2020), contains the following steps with feedback loops: analyze and describe the design problem, set design criteria, develop (partial) solutions, design, build, pilot test, test in practice and evaluate the prototype. In this section, we present the impact goals of the activity and the design criteria. We describe the initial prototype and the methods and results of two subsequent rounds of testing of parts of the concept, and how those led to the final version of the public engagement activity.

### 3.1 Design criteria

In the design of our public engagement activity, we considered the idea of framing (Badullovich et al., 2020). Framing climate change as closer in time, geographically closer, socially closer, and more certain tends to make people more concerned and more inclined to take action to help combat climate change, regardless of how attractive, important, or difficult they see the task (Jones et al., 2016). To reduce that psychological distance, our public engagement activity was designed to emphasize the aspects of sea level rise that are happening in the (relatively) near future, in the Netherlands, to people who are socially similar to them, and that will happen with a relatively high level of certainty. Furthermore, one of the most challenging aspects of

stimulating climate-conscious behavior is the lack of direct feedback (Renes, 2021), so we intended to make an activity where people see more directly how their actions shape the future, with the goal of increasing their *response efficacy*: the belief that their behavior can make a difference in the solution to a problem (Meijers et al., 2018).

Based on these principles, we formulated the following two impact goals for our public engagement activity:

- Positive impact on response efficacy (i.e. 'my actions have influence on sea level rise')
- Positive impact on perceived relevance (i.e. 'sea level rise is relevant to my life')

To evaluate the extent to which the activity has reached these impact goals, it should include an impact assessment, a relatively rare addition to climate communication activities (Wijnen et al., 2023).

In addition to these impact goals, we formulated a number of more general design criteria for the activity. In order to serve as
a science communication effort, the activity should be based on scientific research. The activity should be playful and entertaining, since those aspects seem to be promising for science communication and public engagement with younger audiences (Bättig-Frey et al., 2023). We designed the activity to be suitable for deployment at different occasions, in order to find audiences with higher and lower science capital. The activity should be easy to explain and understand, and should not take longer than 15 minutes.

In summary, these were the design criteria for the activity:
- Including an impact assessment at the end of the activity;
- Based on scientific research;
- Playful and entertaining experience for participants;
- Suitable for a young adult audience, aged 16-25;
- Easy to set up, explain, and understand; and
- Total play time no more than 15 minutes, including instruction and impact measurement.

### 3.2 Developing the initial prototype

Based on the impact goals and design criteria presented in the previous section, we held a brainstorm session with two theatre makers/designers and a sea level rise researcher. The idea of a game was quickly agreed on; partially because it seemed to fit
our young adult audience, and because a choice-based game would enable us to emphasize the importance of participants' actions. While a digital game might be more interesting for a young adult audience, we settled on a board game instead, partially for practical reasons (developing a digital game would quickly exceed our financial and time constraints) and partially for the benefit of having a physical tool that would allow for an easier and more explicit multiplayer experience. The resulting idea was a board game where players could make choices and then see what consequences these would have on their own life.
This fits with the idea of board games as promising tools to stimulate discussion and explain academic research (Whittam & Chow, 2017; Illingworth, 2020).

To make participants realize that sea level rise is relevant to them personally, we chose the year 2080 as an important part of the game, as this is far enough in the future to possibly have a significant amount of sea level rise, while most of the young

adult participants will still be alive by then; this is intended to make the participants feel more connected to the consequences of sea level rise for their personal future. As opposed to the Climate Adaptation Game[1] developed by Swedish Meteorological and Hydrological Institute (SMHI), where players make decisions on a policy level, the dilemmas in the proposed game would be personal, about where players would like to live, how they would prefer to travel and how they want to spend their money. This approach is intended to make the game more interesting and relevant for a younger audience.

In order to make sure that the activity is based on scientific research, we involved a sea level rise researcher in the design process, who emphasized that *mitigation* and *adaptation* are equally important strategies of dealing with sea level rise; *mitigation* means limiting the amount of sea level rise, *adaptation* means implementing measures to deal with (the consequences of) sea level rise (Klein et al., 2007). This insight led to the idea of dividing the players into two teams: one that deals with adaptation measures, another that deals with mitigation measures. Each team makes decisions pertaining to only their part of the strategy, symbolizing the fact that climate policy is always made with incomplete information. Together, the decisions made by the two teams will lead to a certain scenario for the future.

The game was later titled '*Zeespiegelspel 2080*' (Dutch for 'Sea level game 2080') and will be referred to as such in the remainder of this paper.

### 3.3 First design phase: pre-testing the dilemmas

After the idea of the Sea level game 2080 was developed, we wrote the initial version of the dilemmas and resulting future scenarios in consultation with the sea level rise researcher. As a first test, we presented the dilemmas to an audience on the lower end of the target audience's age range. The main goals of this session were to test whether the audience could understand the wording and essence of the dilemmas, whether their choices would be divided more or less equally between the two options of each dilemma, and whether the dilemmas were engaging and relevant for this age group.

### 3.3.1 Test setup & methods

The initial version of the dilemmas was discussed during a 45-minute guest lesson for a class of 19 students of 5 VWO, the fifth (and penultimate) year of university-preparatory secondary education, with students generally aged 17 or 18, putting them on the lower end of our target age group. Due to the limited amount of time, the students were only exposed to six of the eight dilemmas. For each dilemma, the students were asked to choose between the two options in a Wooclap poll[2], and then share their thoughts.

As an example, this is the first of the dilemmas discussed by the students:

*Your sports club is located right next to a dike. The dike needs to be raised, but that also means it needs to be wider, otherwise it won't be stable enough. Your sports club will have to move. Is that OK with you?*

---

[1] https://www.smhi.se/en/climate/education/adaptation-game-1.153788

[2] https://www.wooclap.com/

*A. Yes. With a heavy heart, I vote for merging with our arch-rivals on the other side of town. Their sports field is located in a place where it can certainly remain for the next 50 years. We have to give up our sports field.*

230        *B. No, I do not find that acceptable. We will persuade the city council not to implement this plan. The sports field must remain where it is now and the dike must not be higher.*

### 3.3.2 Outcomes

In general, the students were very engaged with the subject and eager to share their opinions about the presented dilemmas and about climate change and sea level rise in general. There was a lively discussion in the classroom, with students sometimes

actively voicing their frustration at having to make a choice between the two options, and attempting to change each other's minds. The students indicated that most of the dilemmas required considerable deliberation, and the distribution of the answers (Fig. 4) shows that there is some amount of disagreement for each of the dilemmas.

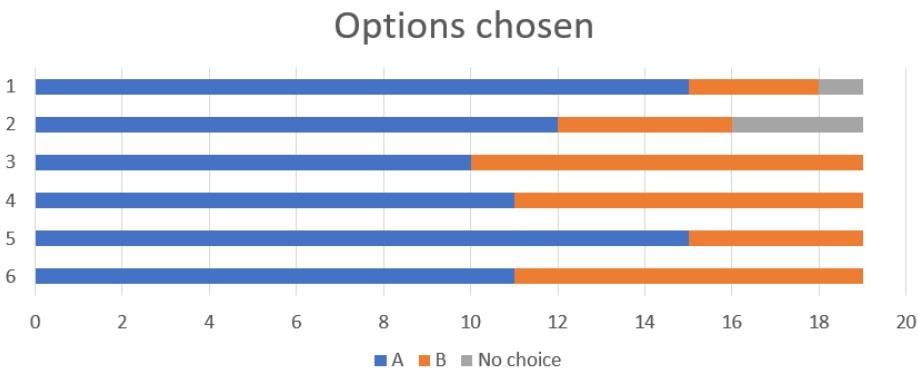

**Figure 4: In our test session with a high school class, we discussed six of the initial dilemmas and asked the 17- and 18-year-old students to choose one of the options. The results show that the choices were divided for all six of the dilemmas.**

Some of the dilemmas did not resonate well with the students' mindset. For example, one of the dilemmas was about a dream job that required an extremely long commute; here, many of the students objected to the term 'dream job', because this commute would be a reason for them to not see the job as such. In one case, where a choice was given between taking a trip

by airplane or by boat, a student even offered a helpful fact check: "What sort of boat do you mean? It only works if you specify that it's a sailboat, because cruise ships are even more polluting than airplanes."

This test session confirmed the suitability of a dilemma-based game for a young audience. Some of the dilemmas were replaced by others, and the wording of the dilemmas was adapted according to the feedback of the test group.

### 3.4 Second design phase: playtesting the game

Since testing a prototype on 'critical friends' is one of the steps in the design process (Veldkamp, 2020), we played the initial version of the complete game, including the new version of the dilemmas, with a group of colleagues. The goal of this session

was to test whether the rules of the game were clear to players, whether it provided an engaging experience, and how much time it took to play the game. Furthermore, we used this gameplay session to test the first version of the impact measurement questionnaire: whether the format and wording of the questionnaire was clear, and how long it would take players to fill it out.

### 3.4.1 Prototype

In the prototype version of the game, each team had their own game leader (as shown in Fig. 5), and the design of the game boards was much more rudimentary (Fig. 6). The dilemmas and future scenarios were different from the final version of the game (see supplementary materials for more details). The initial version of the impact questionnaire consisted of five questions: age, science capital (two representative questions, following Land et al., 2023), response efficacy, and perceived relevance.
The latter four questions were scored on a Likert scale of 1 to 5.

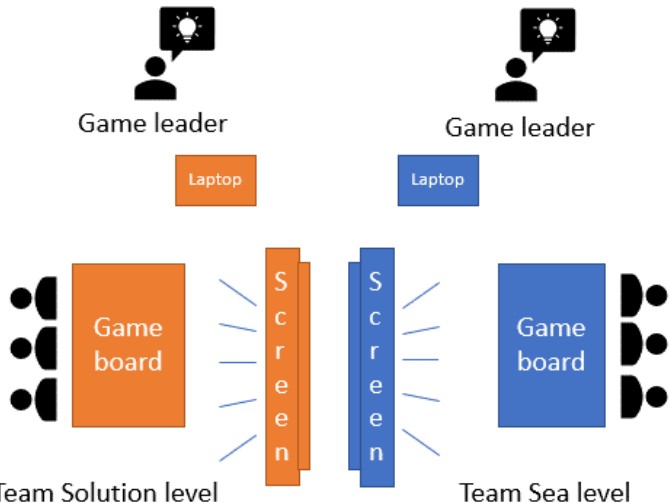

**Figure 5: The first prototype of the Sea level game 2080 had two game leaders, each with their own laptop, and two large computer screens in the middle of the play area, with the game boards and players on either side.**


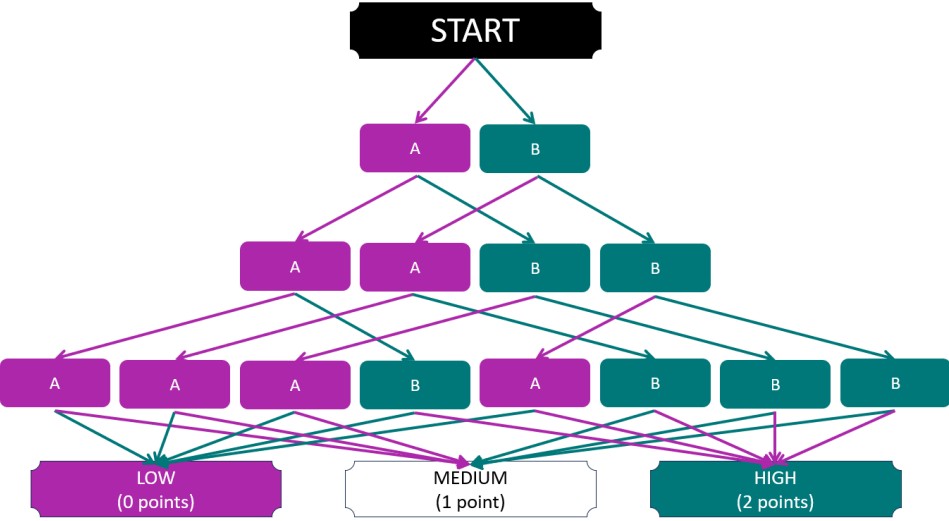

**Figure 6: In the first prototype of the Sea level game 2080, the game board shows the four choices that players make. For each step, players choose A or B; after four dilemmas, players end up in one of three outcomes at the bottom of the game board. For Team Sea level, the three outcomes refer to a low, medium or high amount of sea level rise; for Team Solution level, the three outcomes similarly refer to a low, medium or high amount of mitigation measures being taken.**

### 3.4.2 Test setup & method

The test session was held with a group of colleagues (N = 8), aged 25 to 57. This group is not representative of the target group of this study, but suitable for the goals of this particular test session. After the first play session, the players switched teams and the game was played again.

The lead researcher made observations before, during and after gameplay and recorded these in a text document. No video or audio recordings were made. The participants were informed beforehand that they would be observed, but were asked to play the game as they normally would, in order to make the game experience as natural as possible. After playing the game, there was time for the participants to share their thoughts, experiences and opinions about the game. Some of the participants wrote down additional notes and feedback on the impact questionnaire.

### 3.4.3 Outcomes

Informal observations showed that the game was playable quite easily and quickly, even though there were noticeable variations in the time that participants took to make their choices: some participants made their choices directly after reading through the dilemmas, while others spent several minutes deliberating and had to be urged by the game leaders to make a choice. There was some variation in the choices that the participants made. Some participants expressed difficulty with the loss of nuance that came with the forced choice between two options. For some of the dilemmas, participants pointed out that they found the text of the presented options too long and complicated.

When the participants were presented with the future scenario that resulted from their play session, some indicated surprise about some of the predicted consequences of sea level rise. One participant said: "I thought I knew a lot about climate change, but I had not realized that it would also impact things like the housing problem." Multiple participants experienced the scenarios as quite dystopic: "This is more negative than I thought." "We did our best, and we still ended up in a quite disappointing scenario."

After one participant expressed curiosity about the other scenarios, the participants were presented with some of the alternative scenarios, which helped them realize which consequences their choices had. Still, the participants expressed that they found all the scenarios quite depressing. One participant called it the 'law of conservation of misery': "It makes sense that there is always going to be some amount of trouble. It is interesting to see that it [the type of trouble] is different [in the different scenarios]." Another participant suggested that it might be nice to have at least a little positivity in each of the scenarios, if at all possible.

In the impact measurement questionnaire, the participants had some difficulty answering the two questions pertaining to impact; the Likert scale turned out to be not a good fit in combination with the phrasing of the questions.

Setting up the game took approximately 15 minutes. Explaining the game took about 1 minute. Playing the dilemma phase of the game took 6 minutes in the first session and 10 minutes in the second; the cause of the difference in playtime was not clearly identifiable. In both sessions, reading and discussing the scenarios took 7 minutes. This made the total gameplay time about 15 minutes on average. Filling out the impact questionnaire took 4 minutes.

Based on our findings from the second test phase, we rewrote the dilemmas to be shorter and more concise, and introduced a balance between positive and negative aspects in the future scenarios, as far as possible.

In an attempt to clarify the questions pertaining to impact in the impact measurement questionnaire, the IMPACTLAB *basisinstrument* (Land-Zandstra et al., 2023) suggests using multiple choice options as an alternative to the Likert scale, leading to the following wording of the questions:

*After playing the sea level game…*

       ☐ *I feel much less that my actions affect sea level rise.*

       ☐ *I feel a bit less that my actions affect sea level rise.*

       ☐ *my feelings about this have not changed.*

       ☐ *I feel a bit more that my actions affect sea level rise.*

       ☐ *I feel a lot more that my actions affect sea level rise.*

After consultation with the IMPACTLAB team, we added one more question about the participants' science capital specifically about sea level rise to the impact assessment questionnaire: "I regularly reflect on the consequences of sea level rise." The final version of the science capital questions has a 5-point Likert scale ranging from 'strongly disagree' to 'strongly agree'.

We also made a digital version of the impact questionnaire and added the link as a QR code on the slides with the future scenarios; after finishing the game, participants were given the choice to fill out the questionnaire online or on paper.

Additionally, a number of practical improvements were made to the game, including design adjustments to the game boards, larger game boards, spelling corrections, and playing pieces made of wood instead of plastic.

### 3.4.4 Ethical review

Since the study involved human subjects, we requested an ethical review from the Science-Geosciences Ethics Review Board (ERB) of Utrecht University. Notable aspects in our review request included:

- Our aim to contribute to the current body of research on impact of science communication and public engagement on audiences with different amounts of science capital;
- The low effort for participants, as the project was a simple non-intervention project with a brief standardised questionnaire;
- The caveat that some of the participants would be under the age of 18, since our target audience was adults aged 16 to 25;
- The fact that apart from the participants' age, no personal information was collected, and it would be impossible to trace data back to specific participants;
- Our information and consent letter, which stated the purpose of the study, information on privacy, and contact information for the lead researcher, the secretary of the ERB, and the data protection officer.

The ERB offered a few suggestions to further strengthen our statistical analysis. Furthermore, they requested that we mention in the information and consent letter that participants would be able to quit the survey without any negative consequences and without having to give any motivation for doing so. After we made the suggested changes, the ERB approved our proposal.

**4 Methodology**

### 4.1 Participants

The Sea level game 2080 was designed specifically for a young adult audience, ranging from 16 to 25 years old. While participants of various ages played the game and filled out the impact assessment questionnaire, we only included participants within the 16-25 age range in our analysis. We organized play sessions on occasions where we expected different levels of 345 science capital, in order to gather data from groups both with high and low science capital.

### 4.2 Data collection

The game was played at four different occasions. On 29 September 2023, the Sea level game 2080 was one of 16 'live experiments' at Betweter Festival[3], an annual science and art festival in Utrecht, the Netherlands. The festival is primarily organized by Utrecht University, and connects science, art and society in various ways (e.g., talks, discussions, interviews and

---

[3] https://www.betweterfestival.nl

experiments). In 2023[4], the festival drew 2368 visitors, 4% of which were aged under 21, 31% aged 21-30, 34% aged 31-40, and 31% aged 41 or older. About 89% of visitors indicated that they had completed higher education. During the festival, a total of 106 people played the Sea level game 2080 and filled out the questionnaire, 21 of which were within the target age range of 16 to 25.

On 17 November 2023, the Sea level game 2080 was played at an open day for prospective Bachelor students. The game was located in a building where bachelor's programs from the faculties of Science and Geosciences presented themselves. Attendees of the day were prospective students, in groups or with their parents. The game drew 22 participants, 18 of which were within the target age range of 16 to 25.

On 22 November 2023, the Sea level game 2080 was played with a group of 9 students of the education program '*Leefbare Stad & Klimaat*' ('Livable City & Climate') of a vocational college. All of the students were in the target age range of 16 to 25. Due to logistic difficulties, we played the game online via a Teams video call and a web-based whiteboard, where the students could move their own game pieces in a browser on a large smartboard in the classroom. This worked quite well, and the gameplay experience did not seem to suffer much from these different circumstances.

On 15, 19 and 20 December 2023, we played the game in five first-year classes at a vocational college for media, design and communication. Almost all students were aged 16 to 18, with only a few younger or older. Since the classes were larger than 10 students, we made an extra set of game boards, playing pieces and dilemma sheets, so we could split each class in half and play two games at the same time. The introduction and future scenarios were read out in plenary to the whole class. A total of 71 students participated and filled out the questionnaire, 69 of which fell within the target age range of 16 to 25.

### 4.3 Data analysis

The data analysis section of this paper focuses on evaluating the Sea level game 2080 to determine whether the objectives of this public engagement activity were met. Specifically, the evaluation focused on two key aspects: 1) the impact of the Sea Level Game 2080 on the response efficacy (i.e. 'my actions have influence on sea level rise') and perceived relevance (i.e. 'sea level rise is relevant to my life') among young adults, and 2) whether the science capital predicts the impact outcomes. Descriptive analysis was performed on all items using IBM SPSS Statistics 29 to calculate the means (*M*), standard deviation (*SD*), and frequencies. The differences in mean scores of the science capital measures were examined using repeated-measures ANOVA. Correlations between science capital and impact measures were calculated. A composite science capital score was created based on three science capital items. Cronbach's alpha was computed to evaluate the reliability of the measure, yielding an acceptable value ($\alpha = .68$; Ursachi et al., 2015). We run a simple linear regression analysis for both impact outcome measures.

---

[4] https://www.betweterfestival.nl/rapportage

# 5 Results

**5.1 Statistical results**

A total of 230 responses were collected for the questionnaire. Three participants did not provide consent, leading to their exclusion from the data. Among the remaining respondents, 211 fully completed the questionnaire. Six participants partially filled out 50% of the questionnaire, while 10 participants completed only 17% of the questionnaire. These 16 participants who did not fully complete the questionnaire were excluded from the dataset, as their incomplete responses primarily lacked data

on the main items, specifically impact questions and science capital. A total of 117 participants fully completed the questionnaire, provided consent for participation, and fell within the predetermined age range. The mean and standard deviation values for age, science capital, and impact measures are presented in Table 1.

| | *M (1-5)* | *SD* | 1 | 2 | 3 | 4 | 5 |
|---|---|---|---|---|---|---|---|
| *Age* | 18.65 | 2.94 | | | | | |
| *Science Capital* | | | | | | | |
| (1) I am generally informed about scientific developments. | 3.45 | .91 | – | | | | |
| (2) I regularly discuss science with others at school, at my job or in my free time. | 3.21 | 1.17 | .44** | – | | | |
| (3) I regularly reflect on the consequences of sea level rise. | 2.86 | 1.09 | .36** | .44** | – | | |
| *Impact* | | | | | | | |
| (4) After playing the sea level game, I feel…that my actions affect sea level rise. | 3.44 | .66 | -.03 | .06 | .08 | – | |
| (5) After playing the sea level game, I feel…that sea level rise affects or will affect my life. | 3.57 | .79 | -.16 | .15 | .13 | .38** | – |

\*\* $p < .001$

**Table 1. Descriptive Statistics and Correlations for Science Capital and Impact Variables (N = 117)**

Approximately half of the participants reported being generally informed about scientific developments (51.3%) and engaging in discussions about science regularly, whether at school, work, or during their free time (47%). However, when it comes to regularly reflecting on the consequences of sea level rise, only 34.2% agreed with this statement, with 26.5% expressing

neutrality and 39.4% disagreement. The means of the three science capital measures were not equal [$F(2,232) = 15.4$, $p <$

.001], with the lowest mean score observed in reflecting on the consequences of sea level rise [i.e., "I am generally informed about scientific developments" ($M = 3.45$, $SD = .91$); "I regularly discuss science with others at school, at my job or in my free time" ($M = 3.21$, $SD = 1.17$); "I regularly reflect on the consequences of sea level rise" ($M = 2.86$, $SD = 1.09$)].

After playing the Sea level game 2080, 47% of participants reported an increased feeling that their actions affect sea level rise, while 47.9% remained neutral, with only 5.2% felt less. Moreover, over half (55.6%) reported an increased feeling that sea level rise affects or will affect their lives, with 38.5% expressing neutrality and 6% felt less. Science capital did not predict either of the impact outcomes.

| Variable | B | 95% CI | β | t | p |
|---|---|---|---|---|---|
| Impact – Response efficacy | | | | | |
| (Constant) | 3.30 | [2.82 3.79] | | 13.50 | < .001 |
| Science Capital | .01 | [-.04 .06] | .05 | .56 | .58 |
| Impact – Perceived relevance | | | | | |
| (Constant) | 3.37 | [2.79 3.95] | | 11.52 | < .001 |
| Science Capital | .02 | [-.04 .08] | .07 | .73 | .47 |

Note. $R^2$ adjusted for response efficacy = -.006 and perceived relevance = -.004. CI is confidence interval for B.

**Table 2. Regression analysis summary for science capital and impact outcomes (N = 117)**

In addition to showing that the game positively influenced perceptions about sea level rise, the statistical findings provide valuable insights for future iterations of the activity. The observed lack of correlation between science capital and impact outcomes suggests that the game effectively reaches participants with varying levels of scientific knowledge, making it a versatile tool for public engagement. The science capital score specific to sea level rise is, on average, much lower than the general science capital; this suggests a lack of engagement with the topic of sea level rise, an observation that matches findings from other countries (Jones et al., 2016; Priestley et al., 2021; Leiserowitz et al., 2023; Chilvers, 2014).

**5.2 Observational results**

At Betweter Festival, there was a constant throughput of participants, and the game was often played by the maximum number of people at a time, with new players already waiting in line. Based on anecdotal evidence from observing and speaking to some of the participants, players were highly engaged with the game and the topic. Some of the dilemmas were perceived as much more difficult than others, judging by verbal and non-verbal reactions of players. The game leader's primary responsibility of overseeing the game did not leave sufficient opportunity to formally record the participants' choices and the future scenarios that each game ended up in, but from informal observations it was clear that a large majority of players chose the more climate-conscious options in most of the dilemmas, and a large majority of the games ended up in the future scenarios 'low sea level, high solution level' or 'medium sea level, high solution level'.

At the Bachelor Open Day, there were often relatively small groups of players; many game sessions were played with only one player per team. Participants were enthusiastic and engaged, and indicated that they enjoyed playing a game between taking part in more 'serious' activities such as attending presentations about educational programs. Some of the participants even returned later with a friend to play the game again. In general, there was little discussion after the game sessions, as participants were eager to move on to other activities.

In the online game session with vocational college students, there was relatively little discussion during the game. The participants made very divergent choices, so for both teams, the final outcome was the medium level, which seemed a bit of a letdown for the students. However, the participants became more interested, engaged, and eager to discuss when the game leader presented a number of alternative scenarios. After some discussion about the content of the future scenarios, the conversation turned away from the topic of sea level rise and more towards the design process of the game, which the students

were highly interested in.

     In the offline sessions at a vocational college, the participants tended to make decisions that were not at all climate-conscious. They seemed quite preoccupied by the choices that their classmates made and many of them went along with the majority choice. The students were relatively passive and there was barely any discussion during and after the game. The questions that the students asked the game leader were more basic than in other groups, e.g. "What is the cause of sea level rise?" From their

viewpoint as budding graphic designers, the students did offer some feedback on the design of the game.

## 6 Conclusions and discussion

### 6.1 Results and observations

     In this paper, we describe the design process that led to the Sea level game 2080, a public engagement activity about sea level rise, targeted at a young adult audience. We used an impact assessment to assess the effects on participants' attitudes towards

sea level rise, and whether these effects were correlated with science capital. From our quantitative analysis, we conclude that the Sea Level Game 2080 positively influenced young adults' perceptions regarding the impact of their actions on sea level rise and its effects on their lives. Science capital did not correlate with the impact measures and was not a predictor of the impact outcomes. Among the three science capital measures examined, the one specifically focused on sea level rise received the lowest rating. This might suggest a potential lack of engagement among the participants with topics related to sea level

rise, which is in line with findings from literature about psychological distance to climate change (e.g., Jones et al., 2016).

     These results are consistent with findings from Strick & Helfferich (2022), who observed that science festival activities that focus on personal relevance, interactivity, and accessibility have the strongest positive impact on participants' familiarity with science and scientists and increased knowledge and insight. While Strick & Helfferich's findings focused on participants' attitudes towards science in general, we studied the impact of a public engagement activity about a specific topic. In addition

to a festival setting, we also deployed our activity in an educational setting. Based on this study, we recommend that other

researchers and practitioners incorporate a similar emphasis on personal relevance, interactivity, and accessibility in their public engagement activities, whether focused on a specific topic or on science in general.

Based on the statistical analysis and observations, we conclude that the activity resonated well with our target audience and has a neutral or positive effect on participants' response efficacy and perceived relevance, suggesting that it might lower psychological distance (Jones et al., 2016). The insight that science capital was not a predictor of the impact outcomes may serve as a motivating factor for broadening public engagement efforts to also include groups that are less close to academia (Canfield et al., 2020).

## 6.2 Design process

For the development of our public engagement activity, we followed a design study approach as described in Veldkamp et al. (2020). First, we formulated the impact goals for the activity based on literature about climate communication and public engagement. We based our design criteria on literature and practical considerations about the implementation of the game. The aim to offer participants a playful and entertaining experience quickly led to the idea of a board game, and based on the impact goals for the activity, we chose to develop a dilemma-based game that would show participants the consequences of their choices. Testing the first prototype of the dilemmas with a young adult audience allowed us to make the dilemmas more suitable for this age group. As a second test step, we implemented and tested a prototype version of the complete game, allowing us to finetune the game to better fit the gameplay experience and design criteria.

This iterative nature of the design process enabled us to refine and adapt the activity based on feedback and real-world testing. While the design-based research process used in Veldkamp et al. (2020) was intended for the development of educational materials, we conclude that such an approach is also well-suited for developing a public engagement activity and would recommend that other researchers and practitioners use a similar approach.

## 6.3 Limitations and future directions

The impact assessment focused on analyzing the outputs and outcomes of the public engagement activity. The game was played across four occasions involving a total of 117 young adults. Evaluation was conducted through a brief questionnaire, which included one question about the participants' age, three science capital questions, and two impact questions. While the internal consistency of the science capital scale was deemed acceptable, achieving good reliability, typically around .80, would have been preferable (Ursachi et al., 2015). The impact questions were formulated as "after playing the game" with the aim to evaluate the pre- and post-game change. However, a more comprehensive approach with a split assessment (both pre-test and post-test) might have provided more accurate insights into the game's impact.

Public engagement activities can be evaluated through diverse methods, either as a one-time assessment or after each session. Evaluating each session individually allows for insights that can be used to enhance the overall activity (Reed et al., 2018). Despite the game being pilot-tested before implementation and informal observations being recorded after each session, a more

comprehensive evaluation of the game's design and implementation (e.g., through interviews) could have provided additional insights on how to further improve the effectiveness of the public engagement activity.

It was a challenge to find places to play the game where we expected participants with lower science capital. While we were glad to have finally found two vocational college teachers willing to host us in their classrooms, we do acknowledge that this resulted in a difference between the game sessions in higher and lower science capital settings: the occasions with mostly higher science capital participants (Betweter Festival and the Bachelor Open Day) were more informal and attendees could freely choose whether or not to participate, while at the occasions with mostly lower science capital participants (the vocational colleges), the game was part of a lesson and therefore mandatory. This could explain the observed differences in the participants' enthusiasm, engagement, and willingness to discuss.

Additionally, delving deeper into the decision-making processes of participants during the activity presents an interesting direction for future research. Is there a correlation between science capital and the choices that participants make with respect to sustainability and sea level rise? Also, the vocational college students seemed strongly influenced by their peers' choices in the game; would they make different choices if they kept their choices private? And would that influence the outcome of the impact assessment? Experimenting with different decision-making methods (e.g., keeping players' choices private, limited or no interaction and discussion between players) and recording and analyzing the choices that participants make in relation to their level of science capital might reveal interesting insights. An additional direction for future research could be to evaluate the effect of the activity on participants who are not from the Netherlands, who live further from the ocean, or who live at higher altitudes, and compare this to the effects shown in the current study.

## Data availability

The supplementary materials for this study are available at at https://zenodo.org/records/10931965.

The game is currently being redesigned by students at Grafisch Lyceum Utrecht; when the redesign is finished, the complete game will be made publicly available under a CC BY-NC license.

## Author contributions

NV and EvS conceptualised and planned the study design. NV conducted the design study and the experiments. TV performed the data analysis. NV and TV wrote the article. EvS supervised all steps and commented on the draft versions.

## Competing interests

The authors declare that they have no conflict of interest.

**Ethical statement**

Participation in the Sea level game 2080 was voluntary. Participants were informed beforehand that they were taking part in a scientific study, were able to withdraw without any consequences, and indicated their informed consent on the questionnaire form. The data were gathered and saved without any personally identifiable information. This study was reviewed and approved by the Ethical Review Board of the Faculties of Science and Geosciences of Utrecht University (project number *Bèta S-23129*).

**Acknowledgements**

We thank Sjors & Ruud Theatermakers for their help in designing the game and Tim Hermans for providing us with fact checks. We thank Madelijn Strick for her advice on the impact assessment, and Alice Veldkamp for her informal review of this paper. We thank Nienke de Haan (UniC Utrecht), Joost van Marrewijk (Grafisch Lyceum Utrecht), and Debora Treep (Yuverta Amsterdam) for welcoming us in their classrooms, and Utrecht University for hosting the game at Betweter Festival
and the Bachelor Open Day. Many thanks to our game leaders: Tim van den Akker, Caroline van Calcar, Tim Hermans, Franka Jesse, Arjen Ritzerfeld, Aike Vonk, and Tinder de Waal.

The development of the Sea level game 2080 was partially funded by the Dutch Polar Climate Cryospheric Change Consortium (DP4C) NPP, led by Prof. Roderik van de Wal (Utrecht University).

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
