# Peer review of "Designing and evaluating a public engagement activity about sea level rise"

_EGUsphere, 2024_

## Author Comment (AC1)

Below, we respond (in black) to the comments by reviewer 1 (in blue):

This paper describes a public engagement activity – a game – about sea level rise, aimed at 16–25-year-olds in the Netherlands. The paper shares the design process of the activity through a prototyping stage to development of the final game, and then reports on results of a questionnaire to assess the impact of the game with participants. Public engagement with sea level rise is an important area of research and it is interesting and useful to have insight into the design process that led to the development of this activity, as well as the impact of the game on participants. This will be a valuable paper once some things have been addressed to ensure that the paper is (1) engaging with additional relevant scholarly activity and (2) communicating more clearly with readers.

We thank the reviewer for this positive assessment and agree that these comments will help us further improve the paper.

First, I think the paper should engage with the scholarship on serious games, particularly around coastal adaptation, and situate this study in relation to this work. Here are some papers to look at:

Flood et al, 2018, 'Adaptive and interactive climate futures: systematic review of 'serious games' for engagement and decision-making', *Environ. Res. Lett.* **13** 063005, https://iopscience.iop.org/article/10.1088/1748-9326/aac1c6

Lawrence, J. and Haasnoot, M. (2017). 'What it took to catalyse a transition towards adaptive pathways planning to address climate change uncertainty.' *Environmental Science and Policy*. http://dx.doi.org/10.1016/j.envsci.2016.12.003

Yang, Wei and Harrison, Sarah and Blackett, Paula and Allison, Andrew, 'An explorative analysis of gameplay data based on a serious game of climate adaptation in Aotearoa New Zealand' SSRN, (2024). Available at http://dx.doi.org/10.2139/ssrn.4818597

Lawrence Judy , Stephens Scott , Blackett Paula , Bell Robert G. , Priestley Rebecca, 'Climate Services Transformed: Decision-Making Practice for the Coast in a Changing Climate' *Frontiers in Marine Science*, v8, 2021

https://www.frontiersin.org/journals/marine-science/articles/10.3389/fmars.2021.703902/full

We thank the reviewer for this suggestion and the list of relevant papers, and will add a paragraph in the introduction section where we discuss some of these papers and how our work relates to them.

Other overriding issues are:

- there is too much emphasis in the paper on the prototype, and not enough on the final product. I found it hard to understand the game until I read the supplementary material, and I suggest that some relevant information from the supplementary material be included in the paper

Excellent suggestion; in the revised version of our paper, we will add a section directly after the introduction where we describe the final version of the Sea level game 2080, including the impact questionnaire.

- the order in which information is revealed is not always helpful – some things that would have been useful up front are not revealed until later in the paper

Agreed; the addition of the new section mentioned above should help with this.

- the graphics need better captions, headings and contextualizing

We will amend the captions to better clarify their contents.

- some language needs to be revised for clarity

We will carefully go through the manuscript to improve language, and hope that addressing the specific comments below will solve this issue.

Page specific comments follow:

Abstract: the phrase 'response efficacy and perceived relevance' is not conveying much useful information here, suggest revise

In the revised version of our paper, we will replace this with "reduce participants' psychological distance to sea level rise", which should be clearer for readers.

Line 55: In discussing science capital you should also cite Archer et al 2015 which is in the list of references but is not cited (I suggest a thorough check to ensure papers cited are in the list of references and vice versa)

Thanks for catching this; the Archer et al (2015) reference was of course meant to be cited here, but seems to have disappeared inadvertently. We will add this citation back to this section, and double check the rest as well.

Line 88: what do you mean by 'people like' the participants? Maybe there's another way of phrasing this, referring more specifically to the demographics of this group?

This refers to the concept of psychological distance as introduced earlier in this paragraph; we will amend the text to make clearer what we mean by this.

Line 116: 2080 was chosen as 'most of the young adult participants will still be alive by then' which is a good explanation of why a later date, 2100 for example, was not chosen, but why not an earlier date, say 2050 or 2060 for example? A simple explanation would help.

Good point; we have chosen this year as a point far enough in the future to possibly have a reasonable amount of sea level rise. We will add a few words to explain.

Page 6, figure 1: There's not enough information for this figure to be useful. What do the numbers represent? What are A and B? For this to be useful, the reader needs to know what the six dilemmas are, and how the questions asked relate to this figure. Telling us what the dilemmas are would also help section 2.3 on page 5. If space is an issue, then knowing the specific dilemmas presented to players would be more useful than knowing the options chosen in the prototyping stage.

Agreed; we will update this in the revised version of the paper.

Page 6, 170: was there really a team with only one player? If so, why, and did that impact on how that game went?

As discussed in Section 4.2, there were several sessions where one or both of the teams had only one player. This did not seem to impact the gameplay much, it mostly meant that participants did not have the opportunity to discuss amongst their team and therefore went through the game a bit quicker. We did not record team sizes for the game sessions, but for future research, it might be interesting to see if the lack of discussion in one-person teams has consequences for the game's impact.

Figure 3, p7: For figure 3 to be useful we need to know what 'low' 'medium' and 'high' mean

We will explain this better in the revised version of our manuscript.

Figure 4, p8: Again, we need some information about what 'low' 'medium' and 'high' mean

We will also explain this better in the revised version of our manuscript.

Section 2.5: If there is space, it would be useful to have the final version of the game board, along with the questionnaire, in the paper itself rather than in the supplementary papers. At the moment, there's a real emphasis on the prototype – which is good to learn about the design process – but not enough on the game itself

Absolutely; as mentioned above, we will add a new section with a thorough description of the final version of the game.

Line 288: this phrase 'response efficacy and perceived relevance' is not conveying a lot of information, I suggest rephrase

We will add the descriptions of both terms here, as given in Section 2.1, in our revised manuscript.

Line 330: this line refers to 'low sea level, high solution level' etc – if this is what the low/low, low/medium' etc was referring to in the graphic earlier, this information should be included earlier in the paper

We will explain this better in the revised version of our manuscript.

Line 379: This specific information about the questionnaire would have been useful earlier

This will also be much clearer with the addition of a new section about the final version of the game (including the impact measurement questionnaire), as described above.

Looking at the supplementary papers gave me a much better understanding of the game. If there is space, I would suggest that some specific information about the dilemmas, and the future scenarios, be included in the main paper.

This is a good comment. We will include more specific information about the gameplay (including the dilemmas and future scenarios) in the revised version of the manuscript.

---

## Author Comment (AC2)

Below, we respond (in black) to the comments by reviewer 2 (in blue):

**Overview**

This is an interesting and valuable paper on the design and evaluation of a public engagement activity focusing on sea level rise. The manuscript is very well-written, engaging, and easy to follow. It presents a thoughtful approach to addressing the important topic of climate change, specifically in relation to sea level rise in the Netherlands. There are, however, a few areas where further clarification or expansion could enhance the overall impact of the paper, particularly around the framing of the audience, some methodological choices, and the conclusions.

We thank the reviewer for this positive assessment and agree that these comments will help us further improve the paper.

**Major Comments**

1. **Clarifying the audience in the abstract**. While the abstract is well-crafted, it would benefit from more specific details about the target audience, particularly the Netherlands context. Additionally, it would be helpful to mention the recommendations from your findings and how (or whether) these could be broadly implemented elsewhere, along with any potential limitations.

   We agree that this will improve the clarity of the abstract, and will amend it according to your suggestions.

2. **Section 1.1: Climate change and sea level rise.** The introduction does an effective job of setting the scene, but the rapid transitions between statistics for different countries could be more cohesively presented. Consider linking these statistics together more fluidly and clearly highlighting why the focus of this project is on the Netherlands, addressing the specific challenges faced in this region.

   Thanks for your suggestion. We will add a few sentences to explain why we are referring to studies from other countries (we weren't able to find any recent studies about Dutch attitudes to sea level rise), and clarify that we use these ones from other countries as inspiration, even though the results might not be directly applicable to the Netherlands.

3. **Clarifying 'the public' in Section 1.2**. Throughout the manuscript, there is frequent reference to 'the public.' It would be more accurate to acknowledge the existence of multiple publics, as there is no homogeneous 'general public.' I recommend reworking this terminology throughout. For further context, I discuss this concept in my paper 'A spectrum of geoscience communication: from dissemination to participation' (Illingworth, 2023), but feel free to reference other relevant literature instead. The key point is to ensure you are capturing the diversity of audiences.

   Fully agreed, thanks for catching this. We will update the text accordingly.

4. **Rationale for choosing a board game in Section 2**. The design process is excellent and will be very useful for others working in the field. It would strengthen this section if you explicitly explained why a board game was chosen as the public engagement format over other

possible methods. Additionally, a discussion of the strengths and limitations of this approach, as well as how they were realised in practice, would be valuable.

We thank the reviewer for this comment. In the revised version, we will better explain how we came to the decision of a board game, and the consequences of this choice.

5. **Interpreting statistical analysis in Section 4**. The statistical analysis is well-explained and contextualised, with observational data effectively supporting the analysis. However, I would have liked to see more interpretation of what these results mean, particularly in relation to the 'success' of the game and its potential implications for future development. Expanding on how useful these results were in informing your conclusions would add depth to this section.

While we also interpret the statistical results further in Section 5 (Conclusions and discussion), we agree that some interpretation in this section will also help the reader understand the results better. We will add a paragraph in Section 4 to clarify the statistical results.

6. **Audience biases.** It would be beneficial to reflect on the potential biases or limitations of the audience sampled, for example, acknowledging that participants attending an open day for prospective Bachelor students may have higher levels of science capital and interest compared to other publics. This consideration could help further contextualise your findings.

We discuss this in Section 5.3 (Limitations and future directions).

7. **Conclusions and recommendations**. While Section 5.1 on limitations is particularly strong, the conclusions could be more robust. I would recommend strengthening the final section by tying it back more explicitly to the initial research questions. Including clear, actionable recommendations for other researchers or practitioners based on your work would also be a valuable addition. These recommendations could also be incorporated into the abstract to give the reader a clearer sense of the broader applicability of your findings.

Good suggestion! We will further strengthen the conclusions by tying it back to the research objectives and offering more concrete recommendations, both in this section and in the abstract.

8. **Ethical considerations**. The ethical statement provided is excellent. However, I suggest integrating aspects of this statement into the main body of the text in Section 2, particularly a discussion of the risks and benefits identified during the ethical review process conducted by Utrecht University. This will provide a more holistic view of the study's ethical considerations.

We thank the reviewer for this positive comment and the suggestion to elaborate on the ethical assessment in Section 2, and will do so in our revised version.

**Minor Comments**

1. **Subheadings in Section 1.** While Section 1 provides a very solid introduction, the number of subheadings may make it feel somewhat fragmented. Consider consolidating some of these sub-sections to provide a smoother reading experience.

We will consolidate some of the subsections in the revised version of our paper.

2. **Captions**. The captions are generally clear but could be expanded so that the figures can be interpreted independently of the main text.

We will expand the captions in our revised version of the paper.

3. **Figure 5**. The quality of Figure 5 is somewhat blurry. It might be more effective to present the information as text within the main body of the paper rather than as a figure.

Agreed. We will fix this in our revised version of the paper.

**References**

Illingworth, S.: A spectrum of geoscience communication: from dissemination to participation, Geosci. Commun., 6, 131–139, https://doi.org/10.5194/gc-6-131-2023 , 2023.

---

## Author Response (AR2)

We thank the editor and executive editor for their positive assessment and suggestions for minor revisions.

1. We have replaced "publics" with "audiences".
2. We have added a few words to introduce the name of the game at the suggested location.
3. We have changed the verb in this sentence to plural.